

# Differential gene expression in skeletal organic matrix proteins of scleractinian corals associated with mixed aragonite/calcite skeletons under low $m$Mg/Ca conditions

Ikuko Yuyama[1] and Tomihiko Higuchi[2]

[1] Faculty of Life and Environmental Sciences, University of Tsukuba, Tsukuba, Ibaraki, Japan
[2] Atmosphere and Ocean Research Institute, The University of Tokyo, Kashiwa, Chiba, Japan

## ABSTRACT

Although coral skeletons generally comprise aragonite crystals, changes in the molar Mg/Ca ratio ($m$Mg/Ca) in seawater result in the incorporation of calcite crystals. The formation mechanism of aragonite and calcite crystals in the scleractinian coral *Acropora tenuis* was therefore investigated by RNA-seq analysis, using early growth stage calcite ($m$Mg/Ca = 0.5) and aragonite ($m$Mg/Ca = 5.2)-based corals. As a result, 1,287 genes were up-regulated and 748 down-regulated in calcite-based corals. In particular, sixty-eight skeletogenesis-related genes, such as ectin, galaxin, and skeletal aspartic acid-rich protein, were detected as up-regulated, and six genes, such as uncharacterized skeletal organic matrix protein 5, down-regulated, in low-Mg/Ca conditions. Since the number of down-regulated genes associated with the skeletal organic matrix of aragonite skeletons was much lower than that of up-regulated genes, it is thought that corals actively initiate construction of an aragonite skeleton by the skeletal organic matrix in low-Mg/Ca conditions. In addition, different types of skeletal organic matrix proteins, extracellular matrix proteins and calcium ion binding proteins appeared to change their expression in both calcite-formed and normal corals, suggesting that the composition of these proteins could be a key factor in the selective formation of aragonite or calcite CaCO$_3$.

# INTRODUCTION

Calcium carbonate deposition by scleractinian corals is directly linked to the development of coral reefs, providing the structural basis of high species diversity coral reef ecosystems. Of the two major crystal types of calcium carbonate produced by marine calcifying organisms, aragonite is common in modern scleractinian corals, although calcite occurs in precious corals (genus *Corallium*) as well as foraminifera (*Porter, 2007*; *Porter, 2010*). There is great diversity in calcite versus aragonite spicules in sponges (*Uriz, 2006*), and calcite versus aragonite versus a mix in mollusks (*Belcher et al., 1996*). Although each crystal type is specifically determined by each marine organism, environmental conditions,

Corresponding author
Tomihiko Higuchi,
thiguchi@aori.u-tokyo.ac.jp

such as molar Mg/Ca ratio ($m$Mg/Ca) and seawater temperature, can also influence crystal type selectivity (*Ries, Stanley & Hardie, 2006*; *Higuchi et al., 2017*). Fossil records indicate that Mg concentrations, resulting from fluctuations in past seawater $m$Mg/Ca ratios, have impacted greatly on the selective nucleation of aragonite/calcite crystals (*Ries, 2010*). For example, high $m$Mg/Ca is favorable for precipitating aragonite, and low $m$Mg/Ca, for precipitating calcite (*Balthasar & Cusack, 2015*). During the Cretaceous period, characterized by low $m$Mg/Ca, scleractinian corals (with aragonite skeletons) have been poorly reported form the fossil record whereas rudist bivalves, which produced calcite, were the main reef builder (*Stanley & Hardie, 1998*; *Stanley Jr, 2003*; *Janiszewska et al., 2017*).

Although the mechanisms by which calcite versus aragonite crystals are selectively produced by each marine organism are still incompletely understood, biogenic calcium carbonate is known to contain organic matrices that are key components for skeletal growth and determination of carbonate polymorphism (*Rahman & Oomori, 2009*; *Goffredo et al., 2011*). Changing crystal formation may be accompanied by differing nucleation properties of specific proteins; for example, polyanionic proteins extracted from shells control the crystal phase by switching sequentially between aragonite and calcite (*Belcher et al., 1996*). In corals, galaxin, coral acid-rich proteins (CARPs), and skeletal acidic Asp-rich proteins (SAARPs) have been identified as skeletal organic matrix proteins (*Drake et al., 2013*; *Ramos-Silva et al., 2013*; *Fukuda et al., 2003*). In addition, recent large scale transcriptome and proteomics analyses of corals during initial skeletal formation revealed that the expression of these organic matrix and extracellular matrix-like proteins was prominent in the initial skeleton-building process (*Mass et al., 2016*; *Takeuchi et al., 2016*). The accumulation of such molecular level information promises future discoveries of new molecular mechanisms underlying the crystallization and skeletogenesis of corals.

Previous studies, which demonstrated that juvenile corals produced calcite skeletons when incubated in low-Mg conditions, whereas corals produced 100% aragonite skeletons in ambient conditions (*Higuchi et al., 2014*), helped clarify the skeletal formation process in corals, as well as the aragonite/calcite switching mechanisms. The present study was undertaken to clarify the mechanism of calcite formation in scleractinian corals, by comparing gene differential expression profiles between calcite-formed and normal aragonite-formed corals.

## MATERIALS & METHODS

### Coral specimens

Larval cultures of the scleractinian coral *Acropora tenuis* were obtained from the Akajima Marine Science Laboratory (Okinawa, Japan). Several days after spawning, metamorphosis was induced by exposure of the larvae (in 55 mm diameter plastic containers) to 2 μM Hym-248, as described in *Iwao, Fujisawa & Hatta (2002)*. *Symbiodinium* strains CCMP2556 (clade D), obtained from the Bigelow Laboratory for Ocean Sciences (West Boothbay Harbor, ME, USA), were introduced to *A. tenuis* primary polyps, as described in *Yuyama & Higuchi (2014)*. Juvenile coral polyps were incubated with two different molar

Mg/Ca ratios ($m$Mg/Ca 5.2 and 0.5), each in both natural and manipulated seawater, at 25 °C in thermostatically-controlled incubators (THS030PA; Advantec) with LED lighting (100 μmol m$^{-2}$ s$^{-1}$, 12 h:12 h light:dark cycle). Seawater was changed every 2–3 days. Manipulated seawater ($m$Mg/Ca = 0.5) was prepared by mixing filtered (pore size: 0.22 μm) natural seawater and Mg-free artificial seawater, as described in *Higuchi et al. (2017)*. Four containers (each containing approximately 50 polyps) were prepared, two for natural seawater ($m$Mg/Ca = 5.2) and the other two for low-MgMg seawater ($m$Mg/Ca = 0.5). After incubation for two months, several polyps from each container were treated with NaClO to remove tissue prior to confirmation of their crystal structure, the remaining polyps being fixed in RNAlater (Ambion, Austin, TX, USA) for transcriptome analysis.

## Determination of crystal structure

Crystal structures of the skeleton were determined by X-ray diffraction (XRD). 5 juvenile skeletons produced in $m$Mg/Ca 5.2 treatment and 20 juvenile skeletons produced in $m$Mg/Ca 0.5 treatment were analyzed by X-ray diffractometer (SmartLab, Rigaku, Japan) with a low background silicon holder. Calcite intensity was much stronger than that of aragonite, the specific peak of the latter (<10 wt%) being almost equivalent to the background (as described in *Higuchi et al., 2017*). The presence of aragonite was therefore confirmed by Meigen's stain at 85 °C for 10 min (*Hang, Kato & Wada, 2014*).

## Transcriptome analysis

Two replicates of RNA-seq analysis derived from two containers were prepared for each condition ($m$Mg/Ca ratio 5.2 and 0.5). Each replicate, including 30–40 polyps, was homogenized (Ultra-Turrax T8 Homogenizer; Ika-Werke, Staufen im Breisgau, Germany), and total RNA isolated using a PureLink RNA Mini kit (Life Technologies, Carlsbad, CA, USA) and treated with DNase I (TAKARA, Ohtsu, Japan) to digest genomic DNA. mRNA was then isolated using the NEBNext Poly(A) mRNA Magnetic Isolation Module (NEB, Ipswich, MA), and cDNA libraries prepared using the NEBNext mRNA Library Prep Master Mix Set for Illumina (NEB). Paired-end sequencing of 100 bp was performed by Macrogen Japan (Kyoto, Japan), using a HiSeq 2000 sequencer (Illumina, San Diego, CA, USA). Short reads were first pre-processed, trimming bases with a Phred quality score below Qv = 20 from the 5′ and 3′ ends of each read, and retaining reads ≥ 25 bp. Reads with 30% of bases having Qv ≤ 15 were filtered out using the DDBJ Read Annotation Pipeline (same as *Yuyama et al., 2018*). Sequence data were deposited in *the* DDBJ/EMBL/GenBank databases under accession number DRA007943. The reads were assembled using Trinity v.2.1.1. Transdecoder v.2.1.0 and CD-HIT v. 4.6.1 were used to predict Open reading frames, and filter for redundancy and uniqueness. To isolate coral-derived transcripts, resulting contigs were aligned to coral transcriptome data (including an *A. digitifera* genome (*Shinzato et al., 2011*), and non-symbiotic *A. hyacinsus* and *A. tenuis* transcriptomes from Matzlab (https://matzlab.weebly.com/) and the Center for Information Biology, National Institute of Genetics (DDBJ accession number of IADL01000001-IADL01108246, (*Yuyama et al., 2018*) using BLASTN (*e*-value cutoffs

<1e−15)). Subsequently, trimmed reads were mapped to coral contigs using Bowtie2, and each gene expression level counted using eXpress v.1.5.1. iDEGES/edgeRmethod (*Sun et al., 2013*) was performed to detect differentially expressed genes (FDR <0.05) between normal aragonite coral (grown in natural seawater) and calcite corals (grown in low Mg/Ca seawater). Differentially expressed transcripts were annotated with BLASTX against the (public) UniProtKB/Swiss-Prot protein database (*e*-value cutoffs <1e−6). Gene ontology enrichment analysis was performed using DAVID (https://david.ncifcrf.gov/) (*Huang, Sherman & Lempicki, 2009*) to predict significant biological processes affected by seawater Mg/Ca ratios. Swiss-Prot annotation results of all coral contigs identified here were used to provide a background set for GO enrichment analysis,.

## RESULTS

### Skeletal mineralogy

XRD revealed that the coral skeleton produced in *m* Mg/Ca = 5.2 had an aragonite specific pattern, whereas that produced in *m*Mg/Ca = 0.5 had a calcite specific pattern (Fig. 1). Meigen's stain also indicated a small portion of aragonite in the latter. Accordingly, the two coral types were prepared for RNA-seq, one producing a mixed calcite/aragonite (>90% of calcite and <10% aragonite) skeleton in a low-Mg environment, and the second producing a normal aragonite skeleton in natural conditions (hereafter, the former is described as 'calcite coral' and the latter as 'aragonite coral').

### Differentially expressed genes corresponding to a Mg/Ca ratio change

The RNA-seq analysis resulted in an average of 36.6 million 100-base long reads obtained from each sample. A total of 40,776 coral-derived contigs (mean length 917 bp, N50 length 1,266 bp) were generated by de-novo assembles of all reads and blastn alignment to existing coral data. All reads were then mapped to the 40,776 contigs to detect the expression levels of each contig. Mapping reads were counted as FPKM (Fragments per kilobase of exon per million reads mapped). Comparison of normalized FPKM values between control and low-mg conditions detected 2,035 differentially expressed genes (FDR <0.05), 1,287 being up-regulated and 748 down-regulated in calcite coral. A BLASTX search (*e*-value, 1e−6) of these differentially expressed contigs against the UniProtKB/Swiss-Prot database for gene annotation found 1,143 contigs assigned to known protein sequences, the 50 most significantly up-regulated and down-regulated contigs in calcite coral being shown in Table S1. Putative skeletal organic matrix proteins (galaxin, skeletal aspartic acid-rich protein and uncharacterized skeletal organic matrix protein), extracellular matrix proteins (collagen alpha), and some toxins (toxin AvTX-60A and toxin PsTX-60A) were identified as the most up-regulated genes (Table S1A), with extracellular matrix proteins (dematopontin, tenascin-R and hemicentin-2) and cytochrome P450 being the most down-regulated (Table S1B).

### Gene ontology enrichment analysis

The differentially expressed genes between low-Mg conditions (*m*Mg/Ca = 0.5) and control conditions (*m*Mg/Ca = 5.2) were sorted into enrichment categories, according

(a)

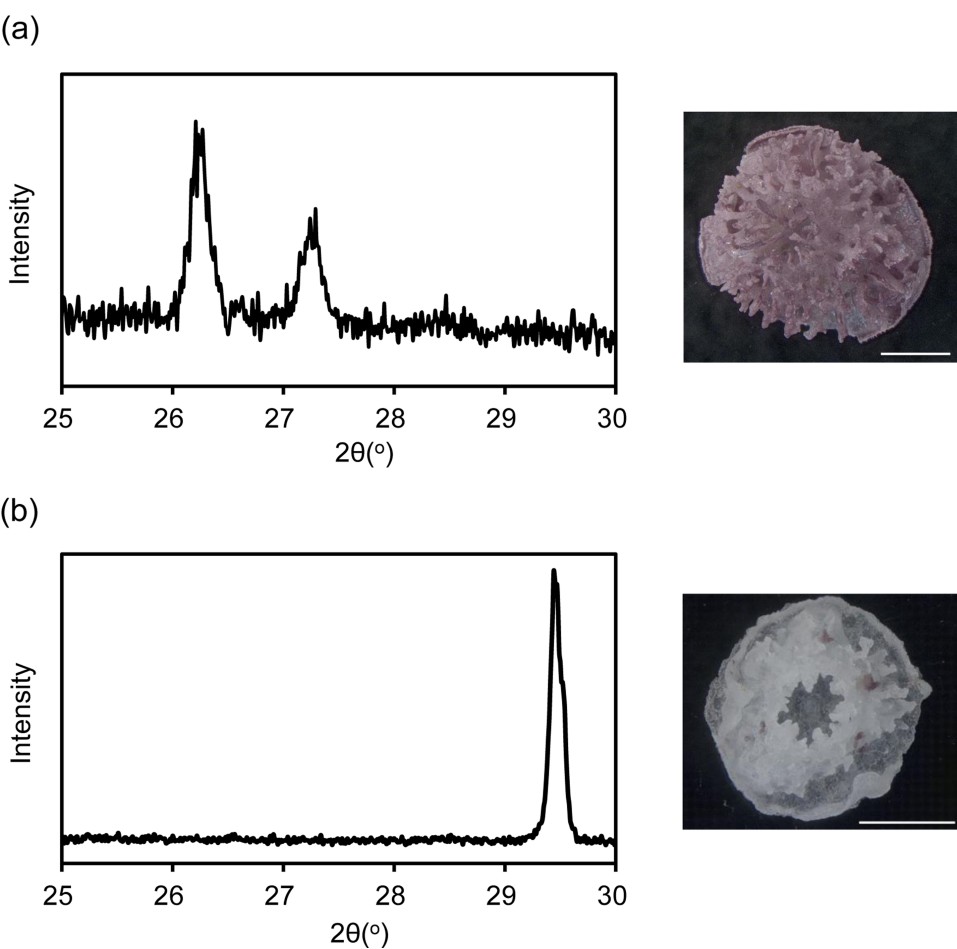

(b)

**Figure 1** **X ray diffraction (XRD) pattern of juvenile skeleton of *Acropora tenuis* and observation image after Meigen's stain.** Aragonite indicated by purple stained regions. (A) $m$ Mg/Ca = 5.2. (B) $m$Mg/Ca = 0.5. Scale bars are 0.5 mm.

to GO enrichment analysis. Significantly enriched gene ontology characterizing the two conditions is described in Fig. 2, with Figs. S1–S10 showing the expression levels and annotation information of contigs contained in each category. The GO analysis indicated that some signaling pathways, for example, Wnt signaling pathway (GO:0016055) and JAK-STAT cascade (GO:0007259), were up-regulated in calcite coral. Moreover, GO terms involved in some metabolic systems, including tetrapyrrole (GO:0046906), secondary alcohol metabolic process (GO:1902652), and extracellular matrix (GO:0031012), were also enriched in up-regulated genes, whereas GO terms involved in stress response [response to oxidative stress (GO:0006979) and necrotic cell death (GO:0070265)] were enriched in down-regulated genes. Figure 2 also showed that GO related to skeletogenesis (calcium ion binding (GO:0005509) and extracellular matrix (GO:0031012)) were detected in both situations, especially in up-regulated genes in calcite coral.

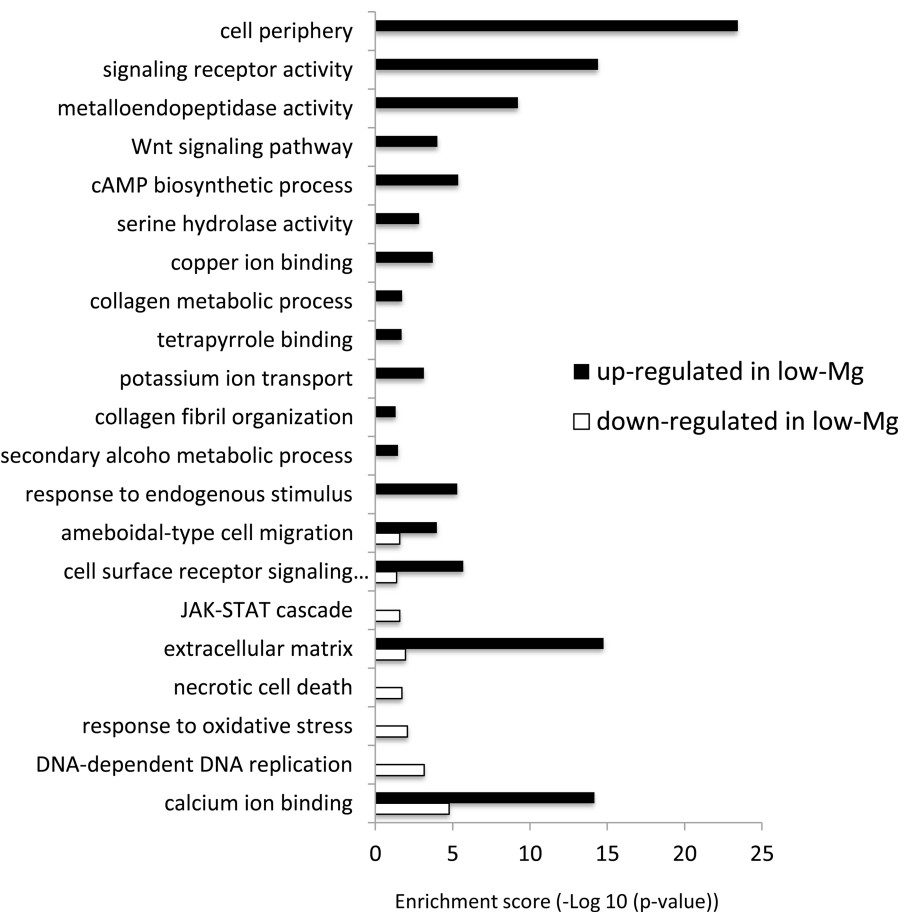

**Figure 2 GO enrichment analysis of differentially expressed genes (DEGs) under normal and low-$m$Mg/Ca conditions.** Vertical axis: GO terms; horizontal axis: enrichment score ($-$Log10 ($p$-value)).

## DISCUSSION

Although scleractinian corals usually produce aragonite skeletons, they can produce calcite–aragonite mixed skeletons (referred as calcite coral) in low-Mg/Ca conditions (*Ries, Stanley & Hardie, 2006*; *Higuchi et al., 2017*). To understand the formation mechanisms of each crystal type in corals, the gene differential expression profile was compared between calcite and aragonite corals. In this study, genes involved in carbonate chemistry of the calcifying fluid, such as Ca ATPase (*Zoccola et al., 2004*) and bicarbonate anion transporter (*Zoccola et al., 2015*), were in neither the top 50 up- or down-regulated genes, nor the full list of DE genes. Thus, we propose that change in $m$Mg/Ca in seawater did not impact the carbonate chemistry of the calcifying fluid, although the calcification rate decreased with low Mg/Ca seawater (*Higuchi et al., 2014*; *Higuchi et al., 2017*). Our transcriptome data for gene subsets related to skeletogenesis is provided in the Tables S2 and S3. These Tables show that upregulation and downregulation of extracellular matrix, acidic protein genes implying the change in expression of these genes contribute to the formation of calcite or aragonite crystal polymorph. Sixty eight potential skeletogenesis-related genes,

i.e., those whose annotation results match skeletal proteins found in *Ramos-Silva et al. (2013)*, were upregulated in calcite corals (low $m$Mg/Ca condition) (Fig. 3). For example, the homologs of galaxin, skeletal aspartic acid-rich protein and uncharacterized skeletal organic matrix protein 2 all showed high expression in calcite corals (Table S2). Moreover, other putative skeletogenesis-related genes, including gene encoding thrombospondin (calcium ion binding protein) (Fig. S5) and toxins, which have been reported as highly expressed genes in the initial steps of coral skeleton formation (*Mass et al., 2016*; *Takeuchi et al., 2016*), were detected. Some toxin types (*Milepora cytotoxin*, toxin AvTX-60A, and toxin PsTX-60A) were markedly elevated (Table S1A), whereas cytotoxin-1 was down-regulated about 90% in calcite coral. Although the role of toxins in coral skeletogenesis is not yet known, AvTX-60A is characterized by cysteine rich repeats (*Bellomio et al., 2009*), being similar to the coral skeletal matrix protein galaxin (*Reyes-Bermudez et al., 2009*; *Fukuda et al., 2003*). In addition, *Ramos-Silva et al. (2013)* reported a further toxin (cephalotoxin-like protein) from the skeleton of *A. millepora*. These findings suggest that some toxins are also included in the coral skeleton organic matrix, and may contribute to skeleton formation. GO enrichment analysis showed many genes involved in "extracellular matrix", "collagen metabolic processes" and "calcium ion binding" increased their expression in calcite corals (Fig. 2). Compared with inorganically precipitated $CaCO_3$ crystals (aragonite/calcite mixed crystals) in similar low Mg conditions, more aragonite forms in coral skeletons (*Balthasar & Cusack, 2015*; *Higuchi et al., 2017*), suggesting that the latter are biologically produced by the expression of genes involved in normal scleractinian coral skeletal formation (i.e., aragonite $CaCO_3$), resulting in the formation of more aragonite crystals even in low $m$Mg/Ca conditions.

It is noteworthy that many genes involved in signal transduction were up-regulated in calcite corals (Fig. 2). For example, the Wnt signaling pathway and JAK-STAT cascade were both identified as up-regulated genes. It has already been reported that the former and TGF-$\beta$/BMP play roles in the biomineralization of corals (*Gutner-Hoch et al., 2017*; *Zoccola et al., 2009*), and these signaling pathways also being involved in collagen secretion in human cells (*Li et al., 2016*). In the present results, the Wnt signaling pathway showed similar expression patterns to the collagen metabolic process, suggesting that it may be involved in coral skeletogenesis through the secretion of collagen. Indeed, collagen related protein has been identified as the skeletal organic matrix of the coral *Stylophora* (*Drake et al., 2013*).

In pearl oyster shells, aspein is an acidic protein which contains a high proportion of aspartic acid (60.4%) in the main body of the protein, which is related to calcite formation (*Tsukamoto, Sarashina & Endo, 2004*). In this study, aspartic and glutamic acid-rich protein were up-regulated as skeletal organic matrix (Table S2). Thus, aspartic acid rich protein may function in calcite formation in scleractinian corals under low $m$Mg/Ca. Down-regulated genes detected in calcite corals during the present study, included some isoforms of dematopontin (Table 1B) [previously identified as a shell organic matrix protein (*Sarashina et al., 2006*; *Jiao et al., 2012*)]. Similarly, the extracellular matrix proteins hemicentin-2 and tenascin-R both had markedly decreased expression (Figs. S3 and S5). Only six genes related to skeletal organic matrix were down-regulated in

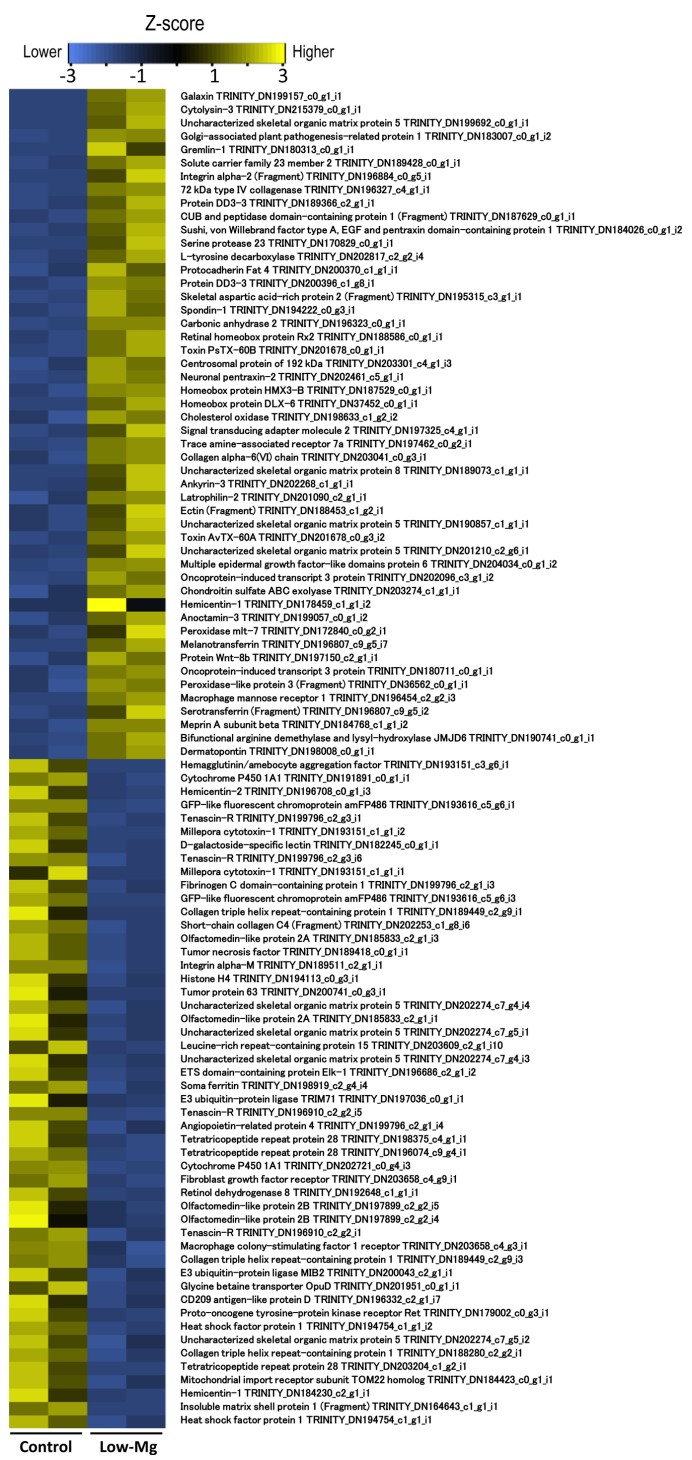

**Figure 3   The top 50 most significantly up-regulated and down-regulated genes.** Colors indicate the expression patterns of each gene (Z-score transformed FPKM values) between control and low-Mg conditions.

the calcite corals (Table S3), whereas many genes coding organic matrix proteins identified from aragonitic corals were up-regulated. It seems likely that even in low Mg/Ca seawater (unfavorable conditions for aragonite production), corals actively promote the skeletal organic matrix to construct an aragonite skeleton more than the calcite alternative. On the other hand, aspartic and glutamic acid-rich protein genes were up-regulated (Table S2). In pearl oyster shells, aspein is an acidic protein which contains a high proportion of aspartic acid (60.4%), which is related to calcite formation (*Tsukamoto, Sarashina & Endo, 2004*). Thus, aspartic acid rich proteins may function in calcite formation in scleractinian corals under low Mg/Ca. Present data suggested that different types of organic matrix proteins, including both skeletal and extracellular, and calcium ion binding proteins were highly expressed in aragonite and calcite corals. It is thought that such variations in expression of organic matrix genes between calcite and aragonite corals may be related to major coral skeleton crystal types.

Stress reactions due to Mg deficiency also occur in calcite corals under low-Mg conditions. Cytochrome P450, known to decrease in expression in Mg-deficient rats (*Becking & Morrison, 1970*), was also downregulated in calcite corals, possibly due to Mg-deficiency. In addition, enriched GO terms associated with "response to DNA-dependent DNA replication (GO:0006261)" in down-regulated genes in low-Mg (Fig. 2), appear to indicate the influence of Mg-deficiency, Mg ions being an essential co-factor for almost all enzymatic system activity acting on DNA processing (*Hartwig, 2001*). The present results also suggested that Mg–deficiency influenced other ion homeostasis; genes related to potassium ion transport (GO:0006813) and copper ion binding (GO:0005507) were up-regulated (Fig. 2), and some heme binding proteins (ferritin and heme-binding protein2) were down-regulated (Fig. 2 and Fig. S6). In particular, a decrease in heme-binding protein may be related to or result from a side effect of calcite formation in corals, since iron ions inhibit the growth of calcite (*Katz et al., 1993*).

## CONCLUSIONS

The present study demonstrated that a large number of genes related to aragonite skeletogenesis in corals were up-regulated under low-Mg conditions. In addition, different types of organic matrix proteins, extracellular matrix proteins and calcium ion binding proteins were expressed in calcite and aragonite corals, suggesting that such proteins might also contribute to coral crystal formation. To clarify whether or not the above gene differential expressions indeed contribute to the formation of calcite or aragonite skeletons, further studies, such as proteome analysis of the skeletal organic matrix in calcite-formed corals, are necessary. Notwithstanding, the present study resulted in a list of candidate molecules involved in biogenetic control of calcite and aragonite formation in coral skeletons, and may also contribute to clarification of the mechanisms of calcium carbonate skeleton formation in various organisms.

## ACKNOWLEDGEMENTS

We are grateful to members of the Akajima Marine Science Laboratory, especially Mr. Kenji Iwao, for provision of *Acropora tenuis* larvae, and also appreciate the members of the Laboratory for DNA data analysis in NIG. RNA-seq data analysis was partially performed on the NIG supercomputer at the National Institute of Genetics. We also appreciate Dr. Graham Hardy for English language editing.

### Funding

This work was supported by a research grant by the Japan Society for the Promotion of Science (#14J40135 and #15K18744 for Ikuko Yuyama, #17H05034 for Tomihiko Higuchi). The funders had no role in study design, data collection and analysis, decision to publish, or preparation of the manuscript.

### Grant Disclosures

The following grant information was disclosed by the authors:
Japan Society for the Promotion of Science: #14J40135, #15K18744, #17H05034.

### Competing Interests

The authors declare there are no competing interests.

### Author Contributions

- Ikuko Yuyama and Tomihiko Higuchi conceived and designed the experiments, performed the experiments, analyzed the data, contributed reagents/materials/analysis tools, prepared figures and/or tables, authored or reviewed drafts of the paper, approved the final draft.

### DNA Deposition

The following information was supplied regarding the deposition of DNA sequences:

The raw FASTQ files for the RNA-seq libraries are available at DDBJ Sequence Read Archive (DRA): DRA007943. http://ddbj.nig.ac.jp/DRASearch/submission?acc=DRA007943.

### Supplemental Information

Supplemental information for this article can be found online at http://dx.doi.org/10.7717/peerj.7241#supplemental-information.

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
