# Peer review of "Differential gene expression in skeletal organic matrix proteins of scleractinian corals associated with mixed aragonite/calcite skeletons under low mMg/Ca conditions"

_PeerJ, doi:10.7717/peerj.7241_

## Round 0.1 · original submission · Major Revisions

Two expert reviewers have evaluated your manuscript and their comments can be seen below and in a downloadable PDF.
Both reviewers have provided important suggestions and observations about this manuscript that requires major modifications. Please ensure that all issues are clearly addressed in a revision.

Reviewer 1 ·

Basic reporting

After revision, the manuscript should be sent to an English grammar service before resubmission.

References should be reviewed for correct spelling, punctuation, and capitalization.

Experimental design

The authors previously (2014) used XRD analysis to quantify calcite versus aragonite in primary skeletons of the same coral species studied in the present manuscript. XRD has the advantage over Meigen’s stain of being unambiguous and quantitative. Since the authors previously performed this analysis (and strengthened their 2014 paper with it), I’m curious as to why this analysis was not performed here. It should be done in place of the Meigen’s stain.

Line 127. DAVID requires a background gene set and there were recently no pre-built ones for corals. Did the authors make their own? If so, they should give details. If they used a pre-built one from another organism, their enrichment could be way off.

Validity of the findings

No comment.

Annotated reviews are not available for download in order to protect the identity of reviewers who chose to remain anonymous.

Reviewer 2 ·

Basic reporting

The current manuscript presents new research pertaining to a coral specie Acropora tenuis. The transcriptomes for Acropora have previously been reported, however, authors of this manuscript tried to investigate the differential expression of genes under low Mg/Ca ratio compared to normal seawater conditions where aragonite is formed by these corals.
Their observations report candidates up-regulated and down-regulated under lowMg/Ca ratio that forms a mixed crystal of calcite and aragonite (which was reported earlier in Higuchi et al. 2017.
Few comments on figures:
Table1: This table has a portion of p and eValues missing in the corresponding columns of the table, and a qualifying cutoff must be added to the figure legend for the convenience of the readers.
Figure1: Some spots on 1a are unstained like 1b, does it mean that 1a also contains calcite? Has this been tested? 1b has some slight stained spots form Meigen’s staining process, with major unstained regions. Has calcite to aragonite ratio been quantified in both the cases? Including this information would be useful.

Experimental design

In this paper, authors are trying to understand the mechanisms involved in making calcite or aragonite in coral skeletons.
Transcriptomics of Acropora tenuis in low Mg conditions showed differential expression of certain genes identified previously.
Adding following information to the manuscript would be helpful.
- A comparison of genes upregulated during calcite formation in shell formers with the current calcite coral data, along with genes upregulated during aragonite formation in aragonite coral vs the previous transcriptomes from Acropora and stylophora. This table would be necessary for a better understanding of differences in calcite and aragonite forming genes.
- A supplemental figure on the list of sequences of proteins mentioned in Figure 3a and 3b from Acropora Tenuis.

Validity of the findings

No comments

Additional comments

No comments

---

## Round 0.2 · Major Revisions

Two expert reviewers have evaluated your revised manuscript and their comments can be seen below. Both agree that substrantial improvements have been made to the manuscript, including further analyses to improve interpretation of the data. However, both also point out that there are issues with grammar that need to be corrected. Also one of the reviewers would like a more thorough re-analysis of the samples by XRD to ensure that all data are of the highest quality, including having high enough XRD counts to make a substantive argument about the relative mineral composition.

Reviewer 1 ·

Basic reporting

I caught a few minor English grammar mistakes that I have taken the liberty of noting in my comments below.

Experimental design

I strongly recommend repeat XRD analysis of the skeleton samples to achieve higher intensities. Normally, the calcite peak is much higher than the aragonite peaks, even when the relative proportion of calcite is very low (as the authors have themselves shown in their previous work with better mineral signal). In the present manuscript, the authors present XRD data where the noise appears to be >10% of the mineral signal peaks. This heavily complicates interpreting mineral relative composition.

Validity of the findings

no comment

Additional comments

I have taken the liberty of making a few suggestions on altered phrasing to improve the English language flow of the text. Added or altered words are in italics.

1. Lines 44, 45. Consider rewording as there is great diversity in calcite versus aragonite spicules in sponges, and calcite versus aragonite versus a mix in mollusks.
2. Line 52. Remove the second ‘mMg/Ca (right before the citation).
3. Line 54. Consider rephrasing as ‘poorly reported form the fossil record whereas rudist bivalves, which produced calcite, were the main reef builder.’
4. Line 56. Consider rephrasing as ‘… by which calcite versus aragonite crystals…’
5. Line 59, 60. Proteins don’t nucleate. Consider rephrasing as ‘Changing crystal formation may be accompanied by differing nucleation properties of specific proteins…’
6. Line 61. Remove the parentheses. They are unnecessary.
7. Line 62, 63. Make CARPs and SAARPs plural.
8. Line 104. Where five juveniles from each treatment analyzed? If so, please explicitly state it.
9. Line 105, 106; Figure 1. I worry that the aragonite signal was not picked up in the updated XRD analysis as either the sample size was too small or the dwell time was insufficiently short. The noise appears as >10% of the signal of aragonite and calcite peaks. Additionally, the authors’ 2014 publication (with very good signal) indicates that under comparable low mMg/Ca conditions, calcite makes up ~20% of the mineral composition rather than >90% for the same coral species. How do the author rationalize this? The authors could easily re-run the samples on a diffractometer with larger sample size and longer dwell time to increase the intensity and allow better interpretation of the relative mineral content.
10. Line 195. Consider rephrasing as ‘Thus, we propose that change in mMg/Ca in seawater…’
11. Lines 200-204. Leave out the part about removing Drake et al. 2013 skeletal proteins from you analysis and consider rephrasing the entire sentence as ‘Sixty eight potential skeletogenesis-related genes, i.e., those whose annotation names match skeletal proteins found in (Ramos-Silva et al., 2013), were upregulated…’
12. Line 207. ‘… including the gene encoding thrombospondin…’ (thrombospondin not capitalized).
13. Line 241. Spell out the number six.

Reviewer 2 ·

Basic reporting

The manuscript has been significantly improved, however,
1. The authors should consider reviewing the manuscript again through a service for English grammar, and especially to form short and crisp sentences.

For example Line 307 in discussion: (The resulting transcriptome data showed different gene subsets related to skeletogenesis.....between calcite and aragonite corals)
can be re-written as:
Our transcriptome data for gene subsets related to skeletogenesis is provided in the Figure-. The figure shows upregulation and downregulation of (x,y,z) genes implying the change in expression of these genes contribute to the formation of calcite or aragonite crystal polymorph.

2. Manuscript title needs to be more accurate (Changing expression in....) to may be differential gene expression in ...low Mg conditions OR Tracking changes in gene expression in...conditions OR Expression of genes change...under ..conditions).

Experimental design

'no comment'

Validity of the findings

Thank you for adding the XRD data and the additional text which has improved the draft.

---

## Round 0.3 · accepted · Accept

I am satisfied with the changes made to the manuscript.

Reviewer 1 ·

Basic reporting

no comment

Experimental design

no comment

Validity of the findings

no comment

Additional comments

Thank you for re-analyzing the 0.5 mM Mg/Ca sample by XRD with a larger sample size so as to obtain higher counts.